# Ciprofloxacin Release and Corrosion Behaviour of a Hybrid PEO/PCL Coating on Mg3Zn0.4Ca Alloy

**DOI:** 10.3390/jfb14020065

**Published:** 2023-01-25

**Authors:** Lara Moreno, Cheng Wang, Sviatlana V. Lamaka, Mikhail L. Zheludkevich, Juan Rodríguez-Hernández, Raul Arrabal, Endzhe Matykina

**Affiliations:** 1Departamento de Ingeniería Química y de Materiales, Facultad de Ciencias Químicas, Universidad Complutense, 28040 Madrid, Spain; 2Institute of Surface Science, Helmholtz-Zentrum Hereon, 21502 Geesthacht, Germany; 3Faculty of Engineering, CAU Kiel University, Kaiserstraße 2, 24143 Kiel, Germany; 4Funcionalización de Polímeros (FUPOL), Instituto de Ciencia y Tecnología de Polímeros (ICTP-CSIC), 28006 Madrid, Spain; 5Unidad Asociada al ICTP, IQM (CSIC), Grupo de Síntesis Orgánica y Bioevaluación, Instituto Pluridisciplinar (UCM), Paseo de Juan XXIII 1, Madrid, 28040, Spain

**Keywords:** corrosion inhibitor, drug delivery, hybrid hierarchical coatings, orthopaedic, magnesium, plasma electrolytic oxidation, polycaprolactone

## Abstract

In the present work, a hybrid hierarchical coating (HHC) system comprising a plasma electrolytic oxidation (PEO) coating and a homogeneously porous structured polycaprolactone (PCL) top-coat layer, loaded with ciprofloxacin (CIP), was developed on Mg3Zn0.4Ca alloy. According to the findings, the HHC system avoided burst release and ensured gradual drug elution (64% over 240 h). The multi-level protection of the magnesium alloy is achieved through sealing of the PEO coating pores by the polymer layer and the inhibiting effect of CIP (up to 74%). The corrosion inhibition effect of HHC and the eluted drug is associated with the formation of insoluble CIP-Me (Mg/Ca) chelates that repair the defects in the HHC and impede the access of corrosive species as corroborated by FTIR spectra, EIS and SEM images after 24 h of immersion. Therefore, CIP participates in an active protection mechanism by interacting with cations coming through the damaged coating.

## Highlights:

Drug eluting hybrid coating comprising a PEO layer and porous polymer top-coat is successfully formed on Mg3Zn0.4Ca alloy.The hierarchical coating system avoids burst release and provides gradual elution of 64% of the drug load over 240 h of immersion.Ciprofloxacin loaded in the coating system shows a corrosion inhibiting effect of 74% compared to non-loaded specimen.Inhibition mechanism of ciprofloxacin involves the formation of insoluble metal chelates with (Ca, Mg) ions released during corrosion from the PEO layer and the substrate that seals the coating defects.

## 1. Introduction

Mg alloys have the remarkable quality of self-degradation and complete resorption in the human body, which makes them an attractive option for use in temporary implants such as cardiovascular stents or orthopaedic implants (screws, bone plates) [1,2]. The preparation of functionalized hybrid coatings, comprising a ceramic base layer and a biodegradable polymer top-coat loaded with the appropriate drug, have been pinpointed as a promising strategy to avoid adverse tissue reactions associated with metallic biodegradable implants while controlling drug release [3]. However, drugs prescribed for clinical reasons could play a major role by either accelerating or eventually delaying implant degradation. In this sense, the inhibiting/accelerating effect may depend on the nature of the drug/coating system. For instance, recent studies on the AZ21 Mg alloy with a PEO coating confirmed the inhibitor role of 8-hydroxiquinoline (8-HQ), while 2,5-pyridincarboxilate (2,5PDC) had the opposite effect when loaded into the PEO coatings [4,5,6]. This was explained by the formation of insoluble and soluble coating-derived species, respectively. Therefore, the drug/coating/substrate compatibility needs to be studied from a wide perspective. 

In the literature, several works have been reported evaluating the potential use of drugs as corrosion inhibitors, although not in the context of bioabsorbable metallic implants. For instance, several drugs have been studied as corrosion inhibitors for mild steel in acidic environments [7,8,9]. For Mg-based implants, to the best of our knowledge, only two reports investigated the effect of drugs such as paracetamol and amoxicillin on the corrosion of AZ91 and AZ31 alloys in physiological solutions [10,11]. They showed that paracetamol and amoxicillin could effectively inhibit corrosion of Mg-Al alloys up to 80–96% due to their adsorption on the alloy surface. The idea of incorporating a corrosion inhibiting drug into coating systems is particularly attractive for Mg alloys given their high reactivity. Indeed, there is much unexplored potential in evaluating the effect of pharmaceutical agents on the corrosion behaviour of Mg alloys. However, the selection of a drug is dictated by its clinical justification and not by the possible effect on corrosion. In cases where accelerated implant degradation is associated with the selected drug, a hybrid hierarchical coating system as a drug carrier may serve as a corrosion impeding solution due to its multi-layer structure. In such a system, a biodegradable polymer top-coat (e.g., polylactic acid, poly(lactic-co-glycolic) acid, poly(ε-caprolactone) (PCL)) can be easily loaded with drugs. PCL is a non-cytotoxic synthetic biodegradable polymer with a relatively long degradation time (~2 years). Its degradation releases protons which can be viewed as a pH equilibrating factor, considering that corrosion of Mg tends to increase the pH [12]. This system would allow for a precision loading of one of the layers with the drug while simultaneously protecting the Mg alloy from corrosion.

The aim of this work is to develop a hybrid ceramic/polymeric hierarchical coating (HHC) system for Mg alloys that incorporates a pharmaceutical agent and to evaluate its release and effects on corrosion behaviour. In more detail, the modification of a Mg3Zn0.4Ca as-cast alloy was carried out by plasma electrolytic oxidation followed by the application of a porous biodegradable PCL layer loaded with ciprofloxacin (CIP). The latter was selected after screening several drugs by hydrogen evolution measurements (results not included here). Drug release was measured by UV-Vis spectroscopy. The corrosion behaviour of the system was studied by means of electrochemical impedance spectroscopy (EIS), scanning vibrating electrode technique (SVET) and microelectrode sensor techniques that probed the local pH and dissolved H_2_ concentration. The corrosion inhibiting mechanism of ciprofloxacin, when forming part of the HHC system, on Mg is discussed. 

## 2. Materials and Methods

### 2.1. Materials

Cast ingot of Mg3Zn0.4Ca alloy (mass fraction: 0.4% Ca, 3.14% Zn, 0.012% Fe, 0.002% Cu and Mg balance) was supplied by the Institute of Surface Science (Helmholtz-Zentrum Hereon, Geesthacht, Germany). The ingot was cut into rectangular specimens (25 × 30 × 4.2 mm^3^) that were ground with SiC abrasive paper to P1200 grit size and cleaned with deionized water and isopropyl alcohol before the PEO treatment.

### 2.2. Coating Procedures

For the PEO treatment, a Ca-P-based electrolyte was developed: 10 g/L Na_3_PO_4_·12H_2_O, 9 g/L Na_2_SiO_3_·5H_2_O, 1 g/L KOH, 2.9 g/L CaO, pH 12.9 and conductivity 22.7 mS/cm. The treatment was carried out using a 2 kW regulated AC power supply (EAC-S2000, ET Systems electronic) in a 2 L thermostated (20 °C) double-walled glass cell, using a 316 L stainless-steel mesh cylinder as counter electrode and an insulated copper wire for the electrical contact. A constant current density of 100 mA/cm^2^ was applied, with input voltage amplitudes of +350/−50 V with 60 s ramp and a PEO treatment time duration of 300 s. After PEO treatment, the specimens were rinsed in deionized water and dried in warm air.

The *Breath figures* (BF) technique consists in formation of a porous polymer layer at high relative humidity (RH > 90%) using a solution of polymer in a volatile solvent. The temperature at the interface decreases when the solvent is evaporated, leading to condensation of vapour water drops on the surface of the sample. As a result, the polymer, solidifying around the drops, forms a porous structure [13]. The BF approach was employed in order to fabricate the porous polymer layer over the PEO-treated Mg. The deposition was carried out by dip-coating method (VT-04 control unit) using PCL as polymer and chloroform as solvent under room temperature and a relative humidity above 80% (monitored by a digital hygro-/thermometer, VWR) in a hermetically sealed chamber. The system designated as HHC consisted of a thin sealing PCL layer and a thick porous PCL top-coat. A sealing layer was applied first onto the PEO in low relative-humidity conditions (<40% RH), using a 0.3 mm/s withdrawal speed, in order to improve the adhesion between PEO and the PCL layer. The porous polymer top-coat was developed inside the closed chamber using 2 mm/s, 2 cycles and 75 mg/mL parameters. For the incorporation of the drug into the porous polymer top-coated layer, 5 wt. % of CIP was dissolved into a PCL/chloroform solution. The coating system loaded with ciprofloxacin was designated as HHC+CIP.

### 2.3. Surface Characterization 

Morphology: the plane view and cross-sectional examination of HHC and HHC+CIP before and after corrosion testing were carried out using a JEOL JSM-6400 equipment operating at 20 kV, which incorporates an OXFORD LINK PENTAFET 6506 EDS analysis system and a backscattered scanning electron detector (BSE). The cross-sections were prepared by grinding to P1200 finish followed by polishing with 3 and 1 diamond pastes. A Nicolet iS50 equipped with a KBr beam splitter and a DTSG-KBr detector and ATR SpectraTech Performer with a diamond crystal was used to perform Fourier transform infrared spectroscopy (FTIR) analysis. Measurements were carried out with 128 scans at a resolution of 4 cm^−1^. The Mg3Zn0.4Ca/PEO specimen was analysed by TEM using a JEOL JEM 2100 instrument operated at 200 kV. The sample was assembled in a sandwich structure (diameter 3 mm) and subsequently thinned until <0.1 mm thickness. After that, the specimen was prepared by ion milling in a 691 GATAN PIPs system with a small incident angle until perforation.

Topography: a focus-variation optical profilometer (Infinite Focus SL, Alicona) and the IF-Measure Suite software were employed in order to analyse the roughness of the samples. The roughness parameters S_a_ (arithmetical mean height of the area) and S_10z_ (ten-point height) were determined from the surface area and the average of three measurements. 

Wettability: contact angle measurements of HHC and HHC+CIP were carried out using a KSV theta goniometer controlling the volume of the droplets to 5.0 μL. A charge-coupled device camera was used to capture the images of the water droplets for the determination of the contact angle values.

### 2.4. Drug Release

Ciprofloxacin release from the HCC + CIP system was measured in the inorganic part of an α-MEM (Eagle minimum essential medium) solution, following the immersion of the specimen (~4.5 cm^2^) in 4 mL of solution at ~37 °C. The solution was prepared in the laboratory using high purity reagents and deionized water (g/L: 6.8 NaCl, 2.2 NaHCO_3_, 0.4 KCl, 0.12 Na_2_HPO_4_, 0.09 MgSO_4_, 0.2 CaCl_2_, pH adjusted to 7.2) and, contrary to commercial α-MEM, it was free of organic additives, therefore, less susceptible to contamination.

At regular time intervals, an aliquot of 2 mL was removed from the testing solution and then, after the measurement, returned into the system in order not to change the concentration of the medium. The drug concentration of the solution was measured using a UV-Vis spectrometer (PerkinElmer instrument, Lambda35) in the range of 200–400 nm in triplicate. Appendix A shows the CIP calibration curve.

### 2.5. Corrosion Testing

All corrosion tests were performed in inorganic α-MEM solution at 37 °C.

#### 2.5.1. Electrochemical Impedance Spectroscopy (EIS)

The electrochemical impedance spectroscopy (EIS) measurements of HCC and HCC + CIP systems were performed using an EChem Analyst (Gamry, USA) potentiostat and a three-electrode cell following 0.5, 1, 2, 3, 6, 24 and 48 h of immersion. The defined area of the working electrode was ~4.5 cm^2^ (1 × 1 × 0.4 cm), the counter and reference electrodes were Pt coil and silver/silver chloride (Ag/AgCl in 3M KCl), respectively. An AC perturbation of 5 mV_rms_ was applied with respect to the open circuit potential (OCP) in the frequency range from 100 kHz to 0.1 Hz. The frequency of 0.1 Hz was chosen to avoid non-stationarity and pseudo-inductive response trigged at low frequencies, and to shorten the measurement time. In order to avoid negative effects of potential non-stationarities, measurements at lower frequencies were avoided. The spectra fitting was carried out using Zview software and ensuring the goodness of fit by means of chi-square values in the range 0.001–0.0001.

#### 2.5.2. Hydrogen Evolution Test

Hydrogen evolution measurements with the HCC and HCC+CIP specimens (~4.5 cm^2^) were carried out during 11 days of immersion. A pH meter connected to a solenoid valve controlled the supply of CO_2_ feeding through the medium in order to maintain its pH at 7.4. This set up is described in more detail elsewhere [14].

#### 2.5.3. Localized Measurements

Corrosion activity of HCC and HCC + CIP systems was studied by monitoring local current densities by scanning vibrating electrode technique (SVET), local pH and local concentration of dissolved H_2_. The working area was delimited with beeswax, exposing a surface area of ca. 4 mm × 4 mm for each sample, unless specified otherwise.

Measurements were performed at 37 °C under hydrodynamic conditions with a flow rate of 1.0 mL/min controlled by a peristaltic pump (Medorex) using a cell volume of 5 mL of inorganic α-MEM solution, thus, renewing the medium every 5 min. The used medium was discarded and not recirculated. 

SVET measurements were performed using an Applicable Electronics equipment with 100 µm distance between the tip and sample surface using a scanned area of 3 mm × 3 mm at a rate of 75 µm per step. The sampling interval at each step was 1 s and the total time for a map (41 × 41 grid) was approximately 1 h, including time for stepwise probe movement. SVET measurements were recorded in mapping mode during 1, 12 and 24 h of immersion. The probe movement and data acquisition were controlled by ASET-LV4 software from Science wares (Inc., Falmouth, MA, USA).

The local pH distribution (mapping mode) was recorded by a commercial glass-type pH microelectrode (Unisense, pH-10), coupled with a mini Ag/AgCl reference electrode. Local H_2_ concentration was measured using an amperometric H_2_ microsensor (H_2_-10, Unisense, Denmark). The readings of both, pH microelectrode and H_2_ microsensor, were controlled by a fx-6 amplifier by Unisense and integrated into a commercial SVET-SIET system (Applicable Electronics) for probe movement. The distance between pH microelectrode and H_2_ microsensor was fixed at 100 µm in horizontal planes using a dual-head stage micromanipulator as illustrated elsewhere [15,16]. The local pH and H_2_ concentration distribution on the surface were scanned at a rate of 100 µm per step and the interval at each step was 3 s. Local pH and local H_2_ concentration measurements were recorded in mode mapping during 1, 6 and 24 h of immersion.

## 3. Results and Discussion

### 3.1. Characterization of Functionalized Materials

Figure 1 shows the SEM images of non-loaded and CIP-loaded porous HHC systems where a homogeneous pore distribution can be seen in both cases. The pores in the CIP-loaded system exhibit a larger average size, 12.2 ± 1.1 µm in diameter (Table 1), compared to the non-loaded specimen (8.5 ± 2.8 µm). The incorporation of CIP occurs in the form of needles penetrating the porous PCL layer (Figure 1b), which may be associated with the reduced solubility of the drug in the polymer PCL/chloroform solution.

In terms of wettability, porous PCL is nearly a hydrophobic surface with a contact angle of 88.9 ± 5.4°. This is explained by the Wenzel model that assumes that the groove under the droplet is filled with air [17]. Further, if the PCL layer of HHC is loaded with CIP, there is only a slight increase in the contact angle of up to 96.1 ± 3.0°. However, this difference cannot be considered statistically significant. According to the literature, CIP is a hydrophilic compound [18], although, clearly, hydrophobicity of the porous PCL surface is maintained in its presence. This can be explained by the pore size and porosity of drug-free and CIP-loaded HHC systems, which were evaluated based on Appendix A micrographs. The S_a_ values were derived from the analysis of the whole area of each 2D micrograph. The incorporation of CIP decreases the number of pores but makes them larger, resulting in a slightly higher pore area fraction (Table 1). 

### 3.2. Controlled Drug Release from Loaded Hybrid Hierarchical Coating

In order to assess the effect of the underlying ceramic layer and the substrate on CIP release, a reference PCL + CIP system, consisting of a series of drug-loaded porous PCL films deposited on a glass wafer (employed as a model substrate) were prepared. A comparison of the drug release from PCL+CIP and HCC+CIP during 240 h of immersion in inorganic α-MEM solution at 37 °C is illustrated in Figure 2. Higher CIP release from porous PCL film on a glass substrate is observed. The loaded PCL film reveals a burst release of ~50% of the entire load during the first 2 h of immersion followed by the release of the remaining load (up to 80–97% of the total) over 48 h. However, in the case of HCC + CIP, a gradual release over time is observed, with only 64% of the load released over the period of 240 h. The possible reason of this may be associated with the fact that Mg^2+^ and Ca^2+^ cations, liberated from the HHC and the substrate, are chelating with CIP^0^ (zwitterionic form, Appendix A) leading to the formation of insoluble chelates in the physiological medium [19] and reducing the CIP release. This observation is in good agreement with other reports where the complexation of different cations present in physiological media with CIP was shown, reducing the bioavailability of the drug [20,21]. However, in the present work it stands to reason that the amounts of Mg^2+^ (0.7 mM) and Ca^2+^ (1.8 mM) initially present in the inorganic part of α-MEM are not enough for significant chelation, since after 240 h in inorganic α-MEM, the CIP release from PCL+CIP film on glass was >80%, while for HCC + CIP it was 64%. This suggests that the predominant sources of Mg^2+^ and Ca^2+^ leading to chelation are the HHC and Mg substrate.

Only a few works have been reported on Mg alloys that were focused on the design of hybrid coatings using polymer nanoparticles as drug-carriers or a monolithic polymer top-coat with dispersed drug-loaded microspheres/nanoparticles [22,23]. The main studied pharmaceutical agents correspond to anti-inflammatories, antibiotics and immunosuppressant agents with the aim to prevent the infection and restenosis avoiding the premature failure of implants [3,24,25]. According to the available literature, a burst release during the first hour of immersion followed by long-term slow release is observed in these systems. In the present work, a direct loading of the polymer layer of HHC is a relatively simple way of functionalization that avoids burst release and ensures a gradual slow release of the load. 

### 3.3. Corrosion Behaviour and Inhibitor Effect of Loaded Hybrid Ceramic-Polymeric Materials

#### 3.3.1. Hydrogen Evolution Test

In order to evaluate the long-term corrosion inhibiting effect of ciprofloxacin on Mg, hydrogen evolution tests were carried out for HHC and HHC+CIP during 11 days of immersion in inorganic α-MEM at 37 °C under flow of CO_2_ at constant pH of 7.4 (Figure 3). 

In Figure 3, the CIP-loaded system shows a steep slope during the first 3 h of immersion, which then decreases and remains relatively constant till the end of the test. The protective effect of CIP-free HCC was maintained only for the first six days, after that period the degradation rate was significantly accelerated, by six times, compared to HCC+CIP. 

The SEM examination of the HHC+CIP system shows the presence of edge defects in the porous PCL layer (Figure 4a). After 24 h of immersion, a crevice at the PCL_sealing_/PEO interface is revealed. Furthermore, the onset of the hydration of the PEO coating and undercoating corrosion products can be seen (as indicated by arrows in Figure 4b). This suggests that during the first hours of immersion, the PCL layer defects at the edges of the sample (Figure 4a) allow for an easy access of the corrosive species through the polymer delamination crevice to the cracks and pores of the outer PEO layer. Further, the TEM examination of the barrier layer of the PEO coating (Figure 5) reveals that the latter is stratified and permeated by plasma microdischarge channels that reduce the true thickness of the barrier layer to <100 nm at some locations. Failure of such a thin barrier layer is what initiates the early undercoating corrosion. This morphology of the inner regions of PEO coatings is in agreement with other studies [15]. 

The protective effect of CIP has been calculated using the measured hydrogen evolution rate (RV_H_) obtained from the expression (RV_H_ = (V_t_ − V_i_)/(T_t_ − T_i_)), where V_t_ and V_i_ are volumes of hydrogen evolved at times T_t_ and T_i_, respectively. The protective effect (η%) is calculated using the expression η% = (RV_H(blank)_ − RV_H(inh)_)/RV_H(blank)_) where RV_H(blank)_ and RV_H(inh)_ are the rates of hydrogen evolution for HHC in the absence and presence of pharmaceutical agent [26]. The result indicates that CIP enhances the protective capacity of HHC by 74%. To date, the effect of CIP was only investigated on corrosion of bare mild steel and bronze, where it exhibited an 86% and 57% inhibiting effect, respectively, in HCl [8,27]. 

#### 3.3.2. Localized Measurements (SVET, Local pH and Dissolved H_2_ Concentration)

Figure 6 illustrates the visual appearances of the specimens and local current density of CIP-free and CIP-loaded HHC systems after 1, 12 and 24 h of immersion in inorganic α-MEM at 37 °C. Localized cathodic activity is developed on the surface of the drug-free coating while predominantly anodic activity is observed in the CIP-loaded coating surface after 1 h of immersion. After 12 h, the current density becomes intensely and overwhelmingly anodic for drug-free HHC (100 μA/cm^2^), while CIP-loaded HHC demonstrates uniformly low current densities that are maintained up to 24 h. The corrosion inhibiting effect of CIP is about an order of magnitude in terms of suppression of corrosion current. In the CIP-free specimen the electrochemical activity gradually localises with time. The formation of more cathodic regions is possibly attributed to the increase in pH by the diffusion of hydroxide ions that occurs at the early stages of corrosion.

On the other hand, considering the concept of H_2_ evolution test, the anodic magnesium oxidation (Mg → Mg^2+^ + 2e^−^) and the cathodic hydrogen reduction (2H_2_O + 2e^−^ → 2OH^−^ + H_2_) occur on Mg alloys’ surfaces. For this reason, the local concentration of dissolved H_2_ and the local pH can be used as indicators of the cathodic reaction. The solubility of H_2_ in water is 1.4 mg/kg at 37 °C and its local concentration in the corrosive solution is affected by the chemical/electrochemical reactions occurring on the surface of Mg during the immersion.

The visual appearance, the variation of local pH and dissolved H_2_ concentration over a period of 24 h for CIP-free and CIP-loaded systems are displayed in Figure 7 and Figure 8, respectively. The pH in CIP-free system within 1 h of immersion quickly increases from 7.2 to 7.95 and then stabilises at 8.33 after 24 h of immersion, demonstrating the uniform electrochemical activity of the sample surface (Figure 7b,e,h). This corresponds to the increasing trend of dissolved H_2_ concentration, from 1.88 µmol/L at 1 h to 2.17 µmol/L after 24 h of immersion. The dissolved H_2_ maps show mostly uniform distribution. 

In the case of the CIP-loaded system, the local pH map reveals a drastic alkalization within 1 h of immersion up to pH 8.33, Figure 8b. However, the pH decreases with time and stabilises around 7.95 after 24 h of immersion (Figure 8e,h). This acidification is in discord with the evolution of dissolved H_2_ concentration, which increases progressively from 3.19 µmol/L within 1 h of immersion to 6.02 µmol/L after 24 h of immersion (Figure 8c,f,i). Notably, localised areas with high amounts of dissolved H_2_ (green and yellow areas, Figure 8f,i) appear after 6 h, which suggest an increased corrosion activity, but do not lead to a pH increase. This apparent disagreement between pH maps and dissolved H_2_ maps may be explained by the continuous CIP release (Figure 9) into the medium. Considering that CIP exhibits two pK_a_ values of 5.9 and 8.89 [28], at pH 7.4 it exists as a ~100% zwitterion. When corrosion of Mg shifts the local pH to 8.3, the equilibrium CIP^0^ (zwitterion) ↔ CIP^−^ (anion) is still shifted to the left (Figure 9). As CIP is being released from the coating, the increase in the zwitterion CIP^0^ fraction reduces the pH [29]. The high release of zwitterionic species and low pH generate more aggressive conditions, where Mg(OH)_2_ is less stable. This tendency should be accelerating the corrosion process, but the opposite happens due to the formation of insoluble Me(II)-CIP chelates (Me = Mg, Ca, mainly released from the Mg/HHC system), which precipitate on the surface of the PEO coating. Therefore, the initial acceleration of Mg corrosion due to weakness of the barrier layer and complexing of the released Mg^2+^ ions is followed by the inhibiting action of the precipitated insoluble chelates, which explains the observed trends in local pH, dissolved H_2_ and current density activity in SVET maps over 24 h.

The cross-sectional SEM examination of HCC and HCC + CIP after 24 h of local pH and dissolved H_2_ concentration measurements (Figure 10) reveals the formation of crevices at the PEO/PCL interface and at the interface of the sealing/porous PCL layer. These crevices lead to the easy penetration of the corrosion species from the electrolyte causing the hydration of the PEO layer (Figure 10a–c) followed by formation of a corrosion products layer under the PEO coating (Figure 10c). In addition, the presence of these crevices accelerates the corrosion process due to high ingress of chloride ions from the medium. This is in agreement with the results of the collected hydrogen volume (Figure 3) and the local electrochemical measurements (Figure 6 and Figure 8).

In the case of the HCC + CIP specimen, the damage at the PEO/substrate interface is negligible compared with the CIP-free specimen, and the coating itself bears lesser signs of hydration. This evidence corroborates the obtained electrochemical and hydrogen evolution results and suggests the formation of insoluble chelates whose precipitation helps to seal the cracks and pores of the PEO coating reducing the degradation rate of Mg. Similar conclusions in regard to inhibiting effect of 8-HQ loaded into a PEO coating on Mg alloy were reached by Vaghefinazari et al. [6]. This is not always the case, as the same authors have shown in [5] that corrosion inhibiting, or accelerating effect of the organic species loaded into PEO coatings will entirely depend on the solubility of the resultant complex compounds. The porous nature of PEO coatings has a capacity to trap and lock the precipitates, resulting in an inhibiting effect. 

FTIR analysis of ciprofloxacin powder, and the drug-free and CIP-loaded HHC systems after 48 h of EIS testing was performed in an attempt to disclose the complex formation (Figure 11). The bands observed in the CIP powder spectrum have good correlation with those reported in the literature for the functional groups of CIP [30], while the spectrum for HHC coincides with that reported for pure PCL [31]. The FTIR spectrum of the HHC+CIP is largely similar to that of the drug-free PCL layer of HHC (Figure 11) and features an additional band at 1103 cm^−1^ corresponding to in-plane C-H vibrations of CIP. Some other changes in comparison with the CIP powder spectrum may be pointing at formation of chelates with the carboxyl and ketone groups. The carboxyl group (-COOH) (the band at 3400–3500 cm^−1^ in the as-received CIP molecule) undergoes deprotonation, due to the substitution of hydrogen by the Mg(II) or Ca(II) ions resulting in the formation of an ionised carboxylate group (COO^−^) in the PCL-CIP sample. As a consequence, the stretching vibration band of the hydroxyl group (υ OH) at 3424 cm^−1^ is reduced, and the stretching band of the carbonyl group (υ C=O, 1591–1612 cm^−1^ present in CIP spectrum) practically disappears in HHC-CIP.

#### 3.3.3. Electrochemical Impedance Spectroscopy (EIS) and Post-Immersion Characterization

Figure 12a illustrates the evolution of the total modulus of impedance at low frequencies (0.1 Hz) for drug-free and CIP-loaded HHC systems. The test was carried out over 48 h of immersion (starting from 30 min) in inorganic α-MEM at 37 °C while maintaining constant agitation. The pH evolution was measured during the test (Figure 12b). 

As it is observed in Figure 12a, the drop in |Z|_0.1Hz_ is of an order of magnitude due to the characteristic delamination of the PCL layer at the edge of the sample. The edges of an orthogonal specimen, where the continuity of a PCL layer may be disrupted due to stress concentration, typically act as the localized corrosion initiation sites (inset macrographs in Figure 12a). The presence of defects at the edges of the specimens allows for an easy access of corrosive species from the medium leading to premature failure of the specimens.

Interestingly, the CIP-loaded system, despite the initial two-times lower |Z|_0.1Hz_ values compared to the drug-free system, demonstrates a recovery of the impedance value after 6 h of immersion. After that, HCC + CIP maintains higher corrosion resistance compared to the non-loaded system. This suggests that CIP presents an inhibiting effect on the corrosion behaviour of PEO-coated Mg. In this work, this effect could be associated with the formation of insoluble chelates such as Mg/Ca-CIP^o^ [19] (Appendix A) that could precipitate in the cracks and pores of the PEO coating blocking the access of the corrosive species of the medium. The complexation of these compounds is predicted to be thermodynamically favourable at neutral pH as it was reported in works of pharmacokinetics of the fluoroquinolone group to which CIP belongs [12,32]. 

Figure 12b displays the evolution of bulk pH during the test. A similar trend is observed for both specimens with a linear increase during the first 6 h of immersion from 7.18 to 8.6. This may be due to the fact that at the beginning the defects at the edges of the sample allow for the easy penetration of the corrosive species from the medium causing the acceleration of degradation rate of the Mg3Zn0.4Ca alloy. After 24 h, both systems exhibited a stabilisation of the pH around 8.8–9.0 due to the formation of corrosion products layer.

The analysis of EIS spectra of drug-loaded and drug-free specimens (Figure 13a) was performed for 1 h and 24 h of immersion in order to evidence the role of chelate precipitates and their contribution to the electrochemical processes. Figure 13a displays the resistances of different parts of CIP-loaded and non-loaded hybrid coatings: response of electrolyte (R_el_), joint response of PCL and outer PEO layer (R_1_), barrier PEO layer (R_2_) and electrochemical activity of the barrier layer/substrate interface (R_3_) in inorganic α-MEM at 37 °C. The Nyquist and Bode plots of the EIS spectra and the equivalent circuit for loaded and non-loaded hybrid coatings are illustrated in Appendix A and the fitting parameters are provided in Appendix A. The results show some pseudo-inductive response at low frequencies, which was attributed to the non-stationary behaviour of the system and was not taken into consideration by the equivalent electric circuit. 

According to Figure 13a, R_1_, R_2_ and R_3_ are higher for the non-loaded system at 1 h of immersion compared to the CIP-loaded one. R_2_ and R_3_ decrease for the drug-free system after 24 h due to the penetration of corrosive species through the edge defects and the cracks in the PCL layer (Figure 13b,c) leading to the hydration of the PEO layer and subsequent degradation of the substrate. The presence of discharge channels in the barrier PEO layer provides additional pathways for corrosive species, causing the degradation of Mg, which is in accordance with the low R_4_ value for the drug-free system after 24 h. 

However, the opposite situation is observed for the CIP-loaded specimen that retains three times higher R_2_ and R_3_ values after 24 h of immersion. These high values are attributed to the formation of insoluble Me(II)-CIP chelates filling the pores and defects of PEO coating, where Me(II) are Mg^2+^ and/or Ca^2+^, easily lixiviated from the PEO layer [33]. This is corroborated by morphological and compositional examination of the specimen surface (Figure 13b–f). The precipitates formed on the HHC specimen surface are discrete Si-rich particles, as amorphous SiO_2_ tends to form and precipitate at neutral pH as a result of lixiviation of Si from the PEO coating [31]. The HHC+CIP system displays lesser cracking compared with HHC (Figure 13b,d). The insoluble chelates are manifested on the HHC+CIP surface as irregular Ca/P containing plate-like crystals with high nitrogen content (point 2, Figure 13e) with nitrogen being the characteristic marker of CIP. It is evident that these precipitated chelate compounds block the passage of the corrosive species through the cracks and pores of the coating system. 

Summarizing, the developed HHC system exhibits a synergetic inhibitory effect, which can be considered as a form of active corrosion control. It is associated with the formation of insoluble CIP-(Mg/Ca) chelates with continuously eluted CIP^0^ zwitterion and cations supplied by the degrading coating and substrate. The chelate precipitates repair the defects in the HHC and impede the access of corrosive species.

## 4. Conclusions

The pore formation on the top-coat porous PCL layer is influenced only slightly by the presence of ciprofloxacin in the polymer/chloroform solution (i.e., pore diameter slightly increases).The drug-loaded HHC system avoided burst release and ensured gradual elution of the load over 240 h. The released ciprofloxacin load fraction from the HHC system on Mg3Zn0.4Ca alloy (64%) was lower than for the PCL film on a glass substrate (>80%) due the complexation of Mg(II) and Ca(II) cations released by the PEO coating and Mg substrate with ciprofloxacin. The drug-loaded HHC system presented overall higher and more stable corrosion protection with time than the drug-free system due to the precipitation of insoluble chelates that impeded the access of corrosive species of the medium through PCL and PEO layers of HHC reducing the degradation rate of Mg.Over 11 days of immersion in pseudo-physiological conditions (at constant pH 7.4 under CO_2_ flow), the CIP-loaded HHC system revealed 74% reduction of the degradation rate of Mg alloy compared to the drug-free one. Local current density distribution maps revealed an order of magnitude corrosion inhibiting effect of ciprofloxacin loaded into HHC-coated Mg3Zn0.4Ca alloy. The active corrosion protection mechanism involves the initial short-term acceleration of Mg corrosion due to the complexing of the released Mg^2+^ ions followed by the coating defect blocking action of the precipitated insoluble chelates of Mg^2+^ and Ca^2+^ cations with continuously released zwitterionic form of ciprofloxacin. 

In conclusion, these results are highly relevant to understanding the influence of drug loading and release on corrosion behaviour of Mg alloys with functionalized hybrid hierarchical coating systems, which is the first step towards the development of customized corrosion protection systems for implants in orthopaedic or cardiovascular applications.

## Figures and Tables

**Figure 1 jfb-14-00065-f001:**
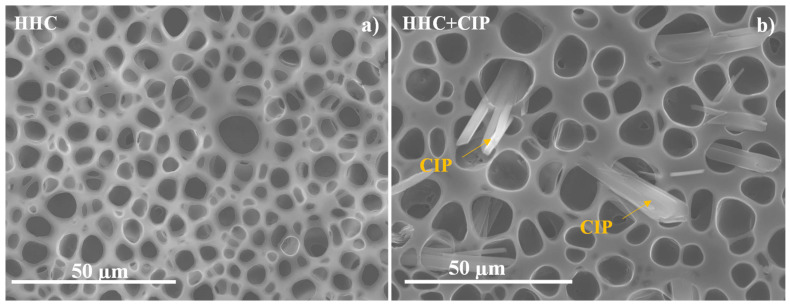
Secondary electron plan view micrographs of HCC and HCC + CIP specimens.

**Figure 2 jfb-14-00065-f002:**
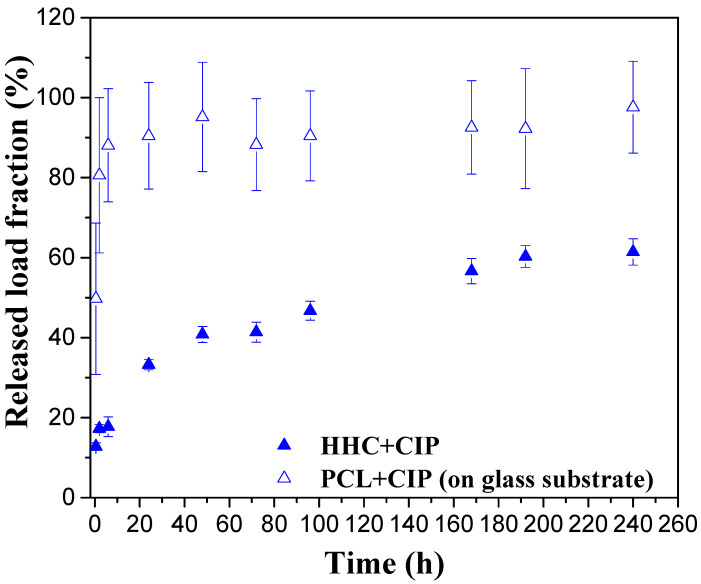
Comparative of released load fraction from porous PCL films and HHC+CIP systems. The porous PCL films were applied onto a model substrate (glass) with 0.57 g of CIP. The HHC+CIP system was loaded with 1.03 mg of the drug.

**Figure 3 jfb-14-00065-f003:**
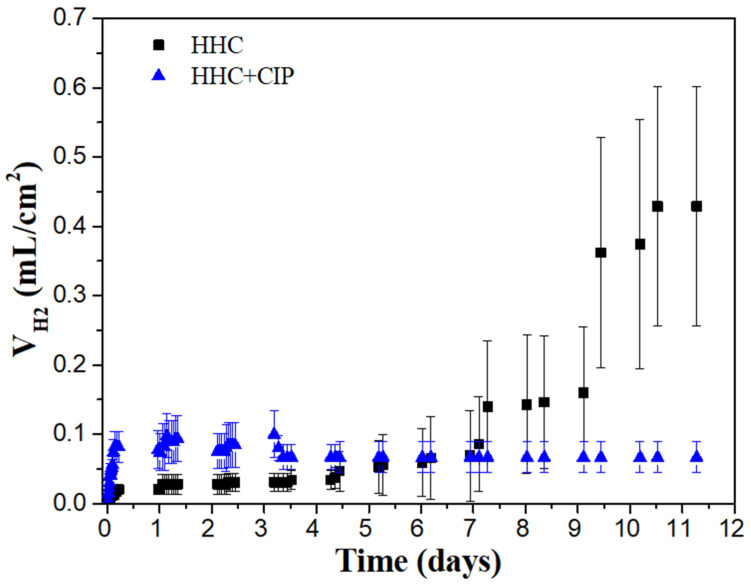
Hydrogen evolution for CIP-free and CIP-loaded HHC on Mg3Zn0.4Ca alloy during immersion in inorganic α-MEM at 37 °C under flow of CO_2_.

**Figure 4 jfb-14-00065-f004:**
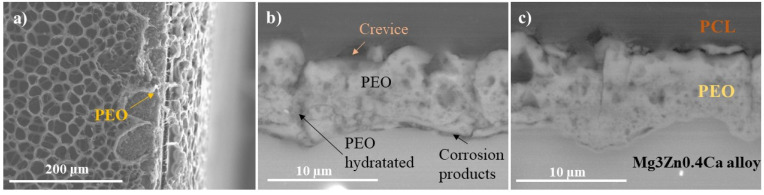
Secondary (**a**) and backscattered (**b**,**c**) electron micrographs of HHC + CIP: (**a**) view of the PCL layer defects at the specimen edge; (**b**,**c**) cross-sectional views following 24 h of immersion in inorganic α-MEM at 37 °C.

**Figure 5 jfb-14-00065-f005:**
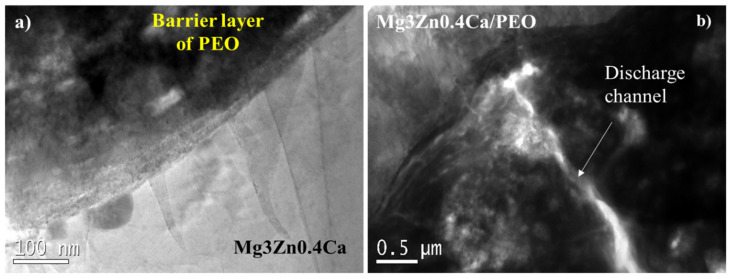
TEM micrographs of the (**a**) barrier layer and (**b**) detail of discharge channel area of the PEO coating on Mg3Zn0.4Ca alloy.

**Figure 6 jfb-14-00065-f006:**
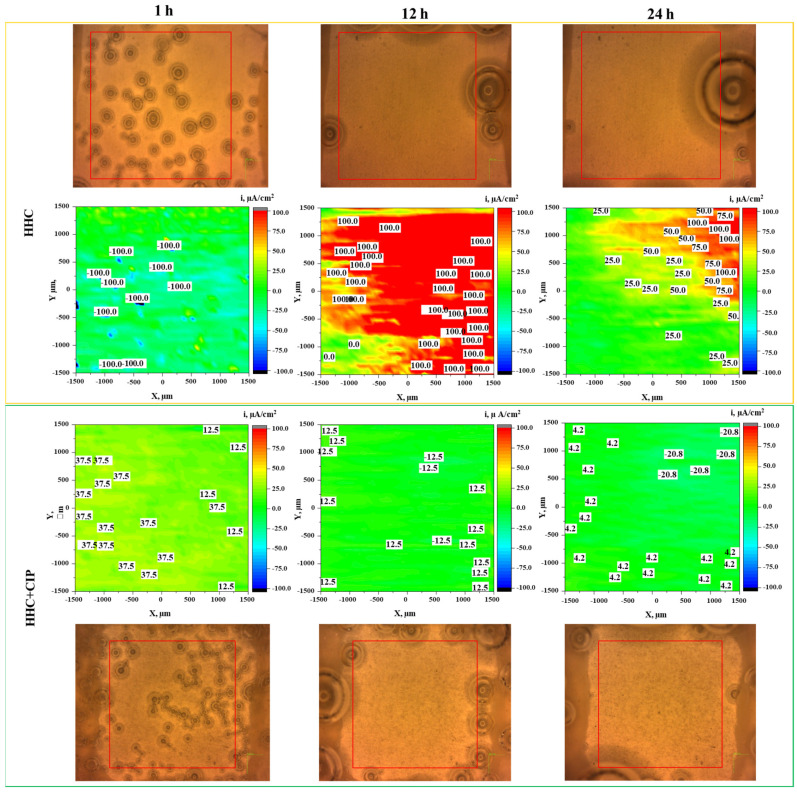
Optical micrographs and SVET current density distribution on CIP-free and CIP-loaded HCC system after 1, 12 and 24 h of immersion in α-MEM at 37 °C. The red rectangle in each optical micrograph indicates the precise location of local pH and H_2_ mapping (3 × 3 mm).

**Figure 7 jfb-14-00065-f007:**
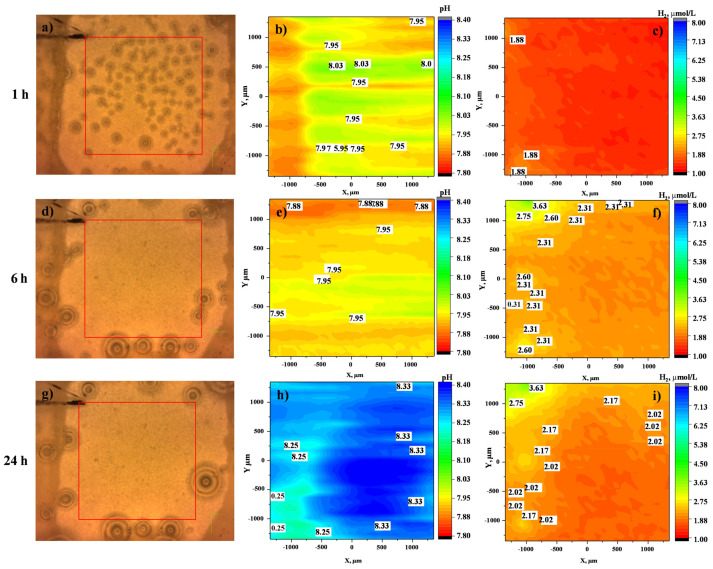
(**a**,**d**,**g**) The visual appearances, (**b**,**c**,**h**) local pH distribution and (**c**,**f**,**i**) local dissolved H_2_ concentration for HHC after 1 h, 6 h and 24 h of immersion in inorganic α-MEM at 37 °C. The red rectangle in each optical micrograph indicates the precise location of local pH and H_2_ mapping (3 × 3 mm).

**Figure 8 jfb-14-00065-f008:**
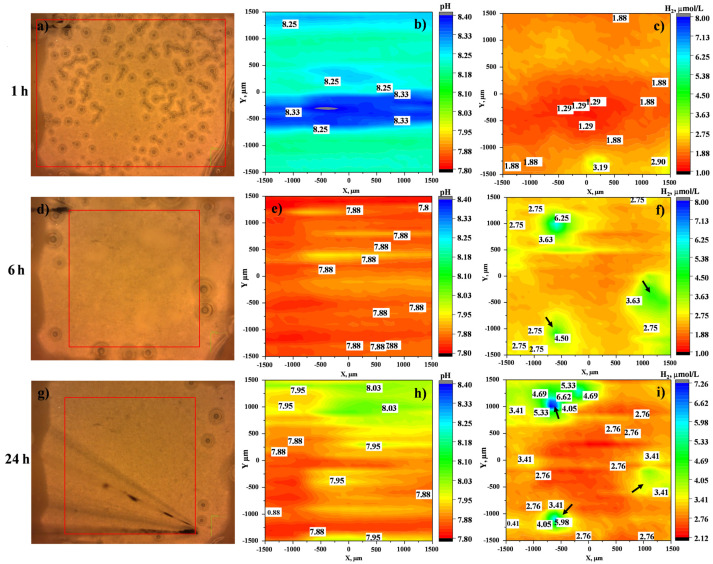
(**a**,**d**,**g**) The visual appearances, (**b**,**c**,**h**) distributions of local pH and (**c**,**f**,**i**) H_2_ concentration of HHC+CIP in inorganic α-MEM at 37 °C after 1, 6 and 24 h. The red rectangle in each optical micrograph indicates the precise location of local pH and H_2_ mapping (3 × 3 mm).

**Figure 9 jfb-14-00065-f009:**
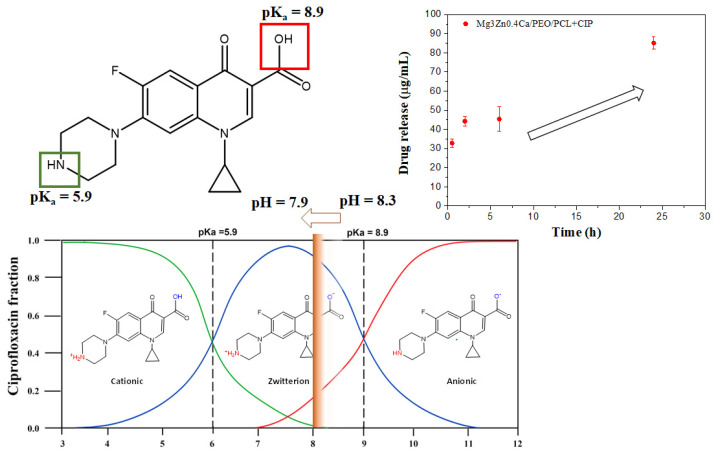
Regulation of pH by CIP zwitterion release.

**Figure 10 jfb-14-00065-f010:**
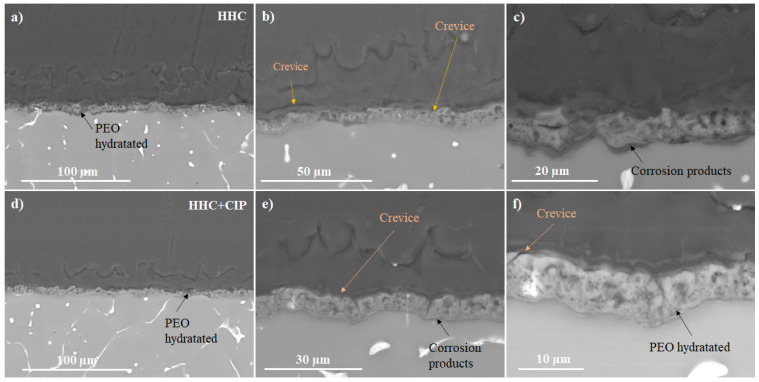
Backscattered electron cross-sectional images of (**a**–**c**) HHC and (**d**–**f**) HHC+CIP after local pH and local H_2_ concentration measurements in inorganic α-MEM at 37 °C.

**Figure 11 jfb-14-00065-f011:**
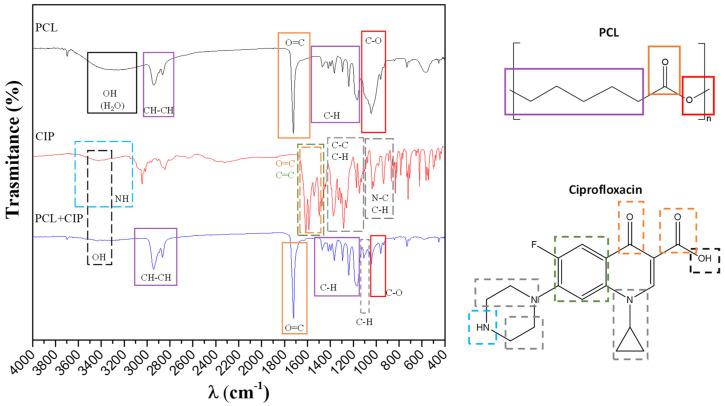
FTIR spectra of ciprofloxacin (red), PCL (black) layer in as-received HHC specimen and PCL + CIP layer in HHC+CIP specimens (blue) after 48 h of immersion in inorganic α-MEM solution.

**Figure 12 jfb-14-00065-f012:**
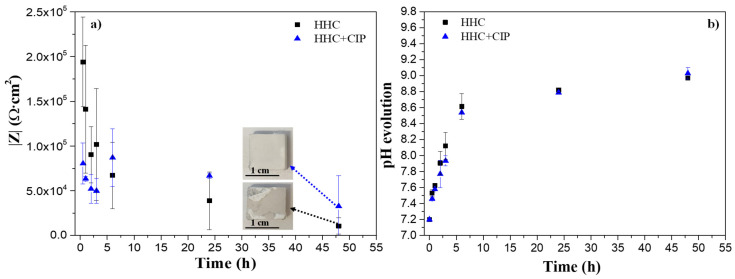
Evolution of (**a**) total modulus of impedance at low frequencies (0.1 Hz) and (**b**) pH for HHC and HHC+CIP systems during immersion in inorganic α-MEM at 37 °C.

**Figure 13 jfb-14-00065-f013:**
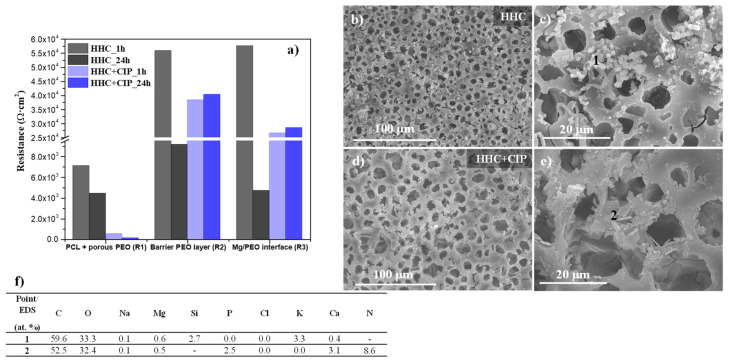
(**a**) The evolution of resistance of the layers of CIP-free and CIP-loaded HHC systems during immersion; (**b**–**e**) secondary electron images of (**b**,**c**) HHC and (**d**,**e**) HHC+CIP specimens after EIS measurements at 48 h; (**f**) local EDS analysis (at. %) of (**c**,**e**) images.

**Table 1 jfb-14-00065-t001:** Surface parameters of drug-free and CIP-loaded hybrid hierarchical coating.

HHC System	Pore Size (µm)	Surface Pore Population Density (Pore/mm^2^)· 10^3^	Contact Angle (º)	S_a_ (µm)	S_10z_ (µm)
HHC	8.5 ± 2.8	298	88.9 ± 5.0	0.6 ± 0.2	7.2 ± 3.7
HHC+CIP	12.2 ± 1.1	76	96.1 ± 3.0	0.5 ± 0.1	5.2 ± 1.5

## Data Availability

The raw/processed data required to reproduce these findings cannot be shared at this time as the data also forms part of an ongoing study.

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
