# Peer review of "Ciprofloxacin Release and Corrosion Behaviour of a Hybrid PEO/PCL Coating on Mg3Zn0.4Ca Alloy"

_jfb, 2023, doi:10.3390/jfb14020065_

Round 1

Reviewer 1 Report

Herein, the author has investigated the Ciprofloxacin release and corrosion behavior of a hybrid 2 PEO/PCL coating on Mg3Zn0.4Ca alloy. I must say that the author has done appreciable and hard work to conduct this research but still, there is room for improvement, thus, I would suggest a minor revision of the manuscript to improve the quality of the article for its possible publication in this reputed journal. Kindly find my comments below:

1.     Highlights must be revised and relevant findings and observations should be included.

2.     The short mechanism mentioned in the abstract should be recast with the aid of insight findings and absorbance on the substrate.

3.     In the current state, the introduction lacks originality and must be expanded with the inclusion of recent literature and must state why there is a need to conduct this experiment and what is the contribution toward society or the industrial sector.

4.     Section 2.3. must be divided into several sub-sections so that the several characterization techniques used can be easily understood with the aid of more discussion on instrumentation.

5.     What is the meaning of in-house section drug release? The author strictly suggested using the proper relevant scientific terminology to explain each variable and study.

6.     Mention the proper dimension of the working electrode with full LXB with exposed area in the electrochemical section.

7.     Why conduct EIS in such low frequency?

8.     Section 2.5.2. need more elaboration.

9.     The fitted circuit diagram must be checked and why there are so many flaws in the Nyquist and bodes plot?

10.  Kindly mention chi-square values, and inhibition efficiency in table S1.

11.  As I can notice there is the inductive effect in the Nyquist plot but there is no discussion of it in the fitting and circuit diagram. Kindly revise and correct.

12.  The author must provide OCP vs. Time plot attained at the time of EIS /PDP experiment conduction.

13.  There are a lot of grammatical errors and typo errors. Kindly check and revise.

14.  Why there is no emphasis given to kinetics study?

15.  References need to be revised and check. The author is suggested to add literature in the article: Coatings 12, no. 10 (2022): 1459, https://doi.org/10.3390/coatings12101459

Author Response

We would like to thank Reviewer #1 since his/her feedback was of great value to improve the interpretation of the results. We hope that the revisions made have improved the quality of the paper, after having assessed and addressed in detail the raised concerns.

Reviewer 2 Report

The aim of the paper, i.e. the ciprofloxacin release and corrosion behaviour of a hybrid PEO/PCL coating on Mg3Zn0.4Ca alloy, is quite interesting and novel.

The abstract summarize the work. The purpose of the study is clearly outlined and the findings of prior work are well discussed.

There are no errors in logic or experimental procedure. The authors accurately explain how the data were collected. There is sufficient information that the experiment can be reproduced.

All topics are very well presented and discussed. The summary and conclusions are sound and justified. All presented figures are very good quality and they prove their point.

The paper is written in good English. The manuscript is easily readable concerning language, style and presentation. The references are appropriate and up to date.

Author Response

We thank the Reviewers for the positive feedback and the appreciation of our work.

Reviewer 3 Report

The manuscript Ciprofloxacin release and corrosion behaviour of a hybrid PEO/PCL coating on Mg3Zn0.4Ca alloy” addresses a hybrid ceramic/polymeric hierarchical coating (HHC) system for Mg alloys. The HHC system incorporates a porous structured polycaprolactone (PCL) top-coat layer, loaded with a pharmaceutical agent (Ciprofloxacin) and this study evaluates its release and effects on the system corrosion behaviour. 

The successful incorporation of Ciprofloxacin as a corrosion inhibiting drug into the coating system is of scientific relevance and possesses merit. Coating characterization is complete. Results are clear and sufficiently discussed and an inhibition mechanism is proposed. 

The paper is well written and is publishable without revisions.

Author Response

(The authors gave the same response as above.)

Reviewer 4 Report

Dear authors,

thank you for submitting your manuscript. 

Some minor comments:

- what are the clinical need and the clinical application of your work? It is not very clear in the text.

- Figures require some improvement. Especially Figures 1, 4 and 13. For Figure 1, resolution needs to be improved, plus adding some square boxes to differentiate subfigures would be beneficial, while some arrows showing the CIP would be good. Resolution to be improved in Figure 4. In Figure 13, the resolution is poor, hence making it difficult to understand that there are six different subfigures.

- Some parts in the text (e.g, in page 14 or 16) are in grey colour. Is there a specific reason for this?

Author Response

We would like to thank Reviewer #4 for the feedback. We hope that the answers and changes in the manuscript provide a satisfactory clarification for the raised issues.
